# Predictive Risk Score for Acute Kidney Injury in Hematopoietic Stem Cell Transplant

**DOI:** 10.3390/cancers15143720

**Published:** 2023-07-22

**Authors:** Natacha Rodrigues, Mariana Fragão-Marques, Cláudia Costa, Carolina Branco, Filipe Marques, Pedro Vasconcelos, Carlos Martins, Adelino Leite-Moreira, José António Lopes

**Affiliations:** 1Division of Nephrology and Renal Transplantation, Centro Hospitalar Universitário Lisboa Norte, EPE, 1649-028 Lisboa, Portugal; claucosta_30@hotmail.com (C.C.); carolinagbranco@hotmail.com (C.B.); filipedcmarques@campus.ul.pt (F.M.); jalopes93@hotmail.com (J.A.L.); 2UnIC@RISE, Department of Surgery and Physiology, Faculty of Medicine, University of Porto, 4099-002 Porto, Portugal; marianaif.rm@gmail.com (M.F.-M.); a.f.leitemoreira@gmail.com (A.L.-M.); 3Division of Hematology, Centro Hospitalar Universitário Lisboa Norte, EPE, 1649-028 Lisboa, Portugal; pedro.de.vasconcelos.monteiro@gmail.com (P.V.); cmartins092@gmail.com (C.M.)

**Keywords:** stem cell transplant, risk factors, predictive risk score

## Abstract

**Simple Summary:**

The incidence and prevalence of hematologic malignancies are increasing throughout the world and hematopoietic stem cell transplant contributes to significantly better outcomes. Acute kidney injury is a frequent complication of hematopoietic stem cell transplants and has known implications for overall survival. We calculated the first simple, easily assessed, inexpensive predictive risk score that helps identify patients with hematologic malignancies undergoing a hematopoietic stem cell transplant at risk for AKI.

**Abstract:**

Hematopoietic stem cell transplant (HSCT) is an important treatment option for hematologic malignancies. Acute kidney injury (AKI) is a common complication in HSCTs and is related to worse outcomes. We aimed to create a predictive risk score for AKI in HSCT considering variables available at the time of the transplant. We performed a retrospective cohort study. AKI was defined by the KDIGO classification using creatinine and urinary output criteria. We used survival analysis with competing events. Continuous variables were dichotomized according to the Liu index. A multivariable analysis was performed with a backward stepwise regression. Harrel’s C-Statistic was used to evaluate the performance of the model. Points were attributed considering the nearest integer of two times each covariate’s hazard ratio. The Liu index was used to establish the optimal cut-off. We included 422 patients undergoing autologous (61.1%) or allogeneic (38.9%) HSCTs for multiple myeloma (33.9%), lymphoma (27.3%), and leukemia (38.8%). AKI cumulative incidence was 59.1%. Variables eligible for the final score were: hematopoietic cell transplant comorbidity index ≥2 (HR: 1.47, 95% CI: 1.08–2.006; *p* = 0.013), chronic kidney disease (HR: 2.10, 95% CI: 1.31–3.36; *p* = 0.002), lymphoma or leukemia (HR: 1.69, 95% CI: 1.26–2.25; *p* < 0.001) and platelet-to-lymphocyte ratio > 171.9 (HR: 1.43, 95% CI: 1.10–1.86; *p* = 0.008). This is the first predictive risk score for AKI in patients undergoing HSCTs and the first study where the platelet-to-lymphocyte ratio is independently associated with AKI.

## 1. Introduction

The incidence and prevalence of hematologic malignancies are increasing throughout the world [1,2]. Several factors contribute to this evidence. Diagnostic standardization according to universal classification systems such as the World Health Organization Classification of Tumour and Haematopoietic and Lymphoid Tissues [3] has resulted in better data collection and higher reporting on regional and international cancer registries. Also, earlier and more precise diagnoses have been a predictable consequence of the evolution of cancer genetic and molecular testing technologies. These two factors have partially overcome underdiagnosis and under-reporting, contributing to a higher incidence of hematologic malignancies. Advanced age is a well-recognized risk factor for cancer development and is partially related to DNA damage accumulation and immunosenescence [4]. The aging of the global population has resulted in a steady increase in hematologic malignancies, both lymphoid and myeloid neoplasms [5]. Higher cancer survivorship rates due to improvements in oncology care have paradoxically created conditions for the development of therapy-related secondary myeloid malignancies [6], which tend to occur in long-term survivors.

With the increasing prevalence of hematologic malignancies, the overall number of patients requiring HSCTs has also evolved. HSCT is a potentially curative treatment for virtually all hematologic cancers and the therapeutic benefits result from high-dose chemotherapy and the graft vs. tumor effect that develops after allografting. Acute leukemias are the most common indications for allogeneic HSCTs [7]. Leukemias harboring adverse genetic events are at high risk of relapse after intensive chemotherapy, and post-remission allografting is the only available treatment modality that can result in a cure or long-term survival [7]. Multiple myeloma is the most common adult indication for autologous HSCTs [7]. The steep dose–response curve for high-dose melphalan in patients with myeloma results in meaningful clinical benefits after HSCT and this treatment modality delays progression and improves median overall survival by approximately 12 months [8]. Relapsed/refractory Hodgkin (HL) and non-Hodgkin lymphomas (NHL) are the second most common adult indications for autologous HSCTs [7] and curability rates are high. The fact that increasing alkylating agent dosing can overcome the resistance of most lymphoma cells defines the biological rationale for the clinical use of autologous HSCT in second-line HL and NHL. In relapsed/refractory diffuse large B cell lymphoma, the most common NHL, the role of HSCTs has been prospectively established and compared to chemotherapy alone; 5-year event-free survival and overall survival were significantly superior in the transplant arm (46% and 53% vs. 12 and 32%, respectively) [9]. In HL, a disease of young adults, autologous HSCT is the standard of care in the relapsed/refractory setting and results in a cure rate of 50% [10].

The number of HSCTs performed has increased 7% per year worldwide from 10,000 HSCTs per year in 1991 to 82,718 first HSCTs per year in 2016, with slightly more autologous (53.5%) than allogeneic HSCTs. HSCT activity has been reported from 87 of the 195 World Health Organization (WHO) member states [11]. More than ever, a better characterization of short- and long-term complications of HSCTs is needed.

Acute kidney injury (AKI) is a possible complication of an HSCT and is known to occur predominantly in the first 100 days after this procedure [12]. The most recent definition of AKI—the Kidney Disease Improving Global Outcome (KDIGO) classification [13]—has been responsible for the uniformization of the various previous definitions in clinical practice, research and public health, contributing to more accurate studies on this matter.

In the last five years, studies have been published on AKI in HSCTs showing an incidence above fifty percent in most studies and a consistent prognostic impact on overall survival, especially in the higher stages of AKI [12,14,15,16,17,18].

AKI still does not have a specific treatment besides treating the cause, which is not always easily identifiable. Prevention and early approach are the best recommendable clinical attitudes [13]. We do consider a predictive risk score for AKI—that can be calculated at hospital admission before undergoing an HSCT for any haemato-oncological diagnosis—to be very useful for the management of these patients.

Our study aims to (1) determine the incidence of AKI in patients undergoing HSCTs using the KDIGO classification with both creatinine and urinary output criteria; (2) to identify independent risk factors for AKI that can be available to the clinician before undergoing an HSCT; and (3) to create a predictive risk score for AKI.

## 2. Materials and Methods

### 2.1. Study Design, Population, and Data Collection

This study was performed considering a single-center retrospective cohort of patients undergoing HSCTs at a tertiary hospital between January 2005 and December 2015. We excluded patients with a previous HSCT, patients with chronic kidney disease already on renal replacement therapy, patients who underwent renal replacement therapy one week before transplantation and patients under the age of 18 years.

The conditioning regimens used followed institutional protocols and none of the patients underwent previous total body irradiation because it is not available at our institution.

Data collection was based on registers of appointments for HSCT eligibility, on registers of the hospital admission period for HSCT (including six-hour period nurses’ records of urinary output and all laboratory analyses performed during this period) and all appointments and hospital admissions in the first 100 days after HSCT.

The following variables were collected: patient’s demographic characteristics (age, gender, race, body weight and height), patient’s comorbidities (diabetes mellitus, hypertension, chronic kidney disease, arrhythmia, valvular heart disease, ischemic heart disease, cerebrovascular disease, chronic liver disease, intestinal inflammatory disease, peptic ulcer, connective tissue disease, chronic obstructive pulmonary disease, solid-organ cancer, psychiatric disease), hematological diagnosis (type of hematological malignancy), transplant´s characteristics (type of transplant, type of donor, cells’ source, induction regimen), laboratory blood panel at admission day (complete hemogram, albumin, uric acid, calcium, phosphate, bilirubin, lactate dehydrogenase, alanine transaminase) and AKI or death during the first 100 days.

### 2.2. Definitions

Chronic kidney disease (CKD) was defined as a persistent decrease in estimated glomerular filtration rate to below 60 mL/min/1.73 m^2^, according to the definition of KDIGO [19].

The hematopoietic cell transplantation-specific comorbidity index (HCT-CI) [20] was calculated according to the latest validated version considering patients’ comorbidities.

Body mass index (BMI) was calculated by dividing the patient’s weight in kilograms by the square of their height in meters.

Platelet-to-lymphocyte ratio was calculated by dividing platelet count by lymphocyte count.

Baseline glomerular filtration rate was estimated according to the Chronic Kidney Disease Epidemiology Collaboration (CKD-EPI) equation [21], considering serum creatinine at the last medical appointment before hospital admission for HSCT as baseline serum creatinine.

AKI diagnosis was made based on daily values of serum creatinine and 6 h urinary output until hospital discharge and all other hospital admissions or weekly evaluations at an outpatient clinic for the first 100 days after undergoing HSCT. AKI was defined by the KDIGO criteria [5] (any of the following: increase in serum creatinine by ≥0.3 mg/dL (≥26.5 µmol/L) within 48 h; increase in serum creatinine to ≥1.5 times baseline, which is known or presumed to have occurred within the prior 7 days; or urinary output <0.5 mL/kg/h for 6 h).

### 2.3. Statistical Methods

Categorical variables were described as frequencies, continuous variables with normal distribution were expressed as mean and standard deviation (SD), and other continuous variables were expressed as median and interquartile range (P25 = 25th percentile; P75 = 75th percentile).

We followed the statistical methodology suggested by the European Group for Blood and Marrow Transplantation [22]. We used survival analysis methods considering competing events—we considered death a competing risk event—through the Fine and Gray method [23] to calculate the cumulative incidence of AKI and perform univariable and multivariable analyses of factors predicting AKI. To establish the multivariable model for the creation of the clinical score, continuous variables that presented a *p* < 0.2 were dichotomized according to the Liu index. The resulting categorical variables were tested for their association with the risk of incident AKI, and a multivariable analysis was performed with a backward stepwise regression (entry criteria—*p* < 0.2). Harrel’s C-Statistic was used to evaluate the performance of the model, and a risk score was created by attributing points corresponding to the nearest integer of two times each covariate’s hazard ratio. The Liu index was used further to establish the optimal cut-off for the clinical risk score, and a log-rank test and respective reverse Kaplan–Meier curves were used to compare the event distributions of the resulting score categorical variable.

Missing data for all variables represented less than 10%; therefore, no imputation techniques were used. Analyses were performed with the statistical software package STATA 16.0 for Windows.

## 3. Results

Five hundred and thirty-four patients underwent HSCT at our center between January 2005 and December 2015. Among these patients, one hundred and twelve patients were excluded for presenting at least one exclusion criteria and 422 patients were eligible for the study.

Patients’ baseline characteristics and transplant-related aspects are shown in Table 1.

### 3.1. AKI—Cumulative Incidence

The AKI cumulative incidence was 59.1% at 100 days after HSCT (Figure 1).

### 3.2. Variable Analysis and Predictive Score for AKI

In the univariable analysis performed for AKI, considering death as a competing event, the variables associated with AKI in this analysis were HCT-CI ≥ 2, chronic kidney disease, hematologic diagnosis—leukemia or lymphoma, basal eGFR, leukocytes count at admission, lymphocytes count at admission, reactive C protein at admission, lactate dehydrogenase at admission and platelet-to-lymphocyte ratio at admission. Each variable with its respective hazard ratio, confidence interval and *p*-value is shown in Table 2.

Considering only variables of Table 2 with *p* < 0.200 and applying the Liu index to establish the optimal cut-off point for each variable, the variables associated with AKI in the univariable analysis were HCT-CI ≥ 2, chronic kidney disease and hematologic diagnosis—leukemia or lymphoma. Each variable with its respective hazard ratio, confidence interval and *p*-value are presented in Table 3.

In multivariable analysis, the variables with independent association with AKI were HCT-CI ≥ 2 (HR: 1.47; 95% CI: 1.08–2.00; *p* = 0.013), chronic kidney disease (HR: 2.10; 95% CI: 1.31–3.36; *p* = 0.002), hematologic diagnosis—leukemia or lymphoma (HR: 1.69; 95% CI: 1.26–2.25; *p* < 0.001) and platelet-to-lymphocyte ratio > 171.9 (HR: 1.43; 95% CI: 1.10–1.86; *p* = 0.008) (Table 4).

The attributable score points to each variable are shown in Table 4. AKI cumulative distribution by score is shown in Figure 2. For a score >3, there was a higher unadjusted risk of incident AKI at 100 days of follow-up (log-rank < 0.001), with an AKI probability of 75.6% [95% CI 82–68.7%] (N day 0 = 227, N day 100 = 107), while patients with a score of 0–3 presented a probability of 47.2% [40.5–53.5%] (N day 0 = 156, N day 100 = 38).

## 4. Discussion

The wide range of AKI incidence in HSCTs amongst older studies—from 20% to 92% [24]—has been explained in part by the use of different AKI definitions. The most recent definition of AKI, the KDIGO classification [13], was proposed in 2012. It resulted from the fusion of the former classifications Risk, Injury, Failure, Loss of kidney function, End-stage kidney disease (RIFLE) [25] in 2004 and the Acute Kidney Injury Network (AKIN) [26] and has been used worldwide aiming the uniformization of this concept in the scientific community.

In our study, 59.1% of patients undergoing HSCTs developed AKI as defined by the KDIGO classification considering both creatinine and urinary output criteria in the first 100 days after HSCT. This cumulative incidence is in the range of recent studies (49.2–68.9%) that consider KDIGO classification—either by creatinine criteria alone [12,14,18] or by both creatinine and urinary output criteria [15,16] in populations with different hematologic malignancies undergoing HSCTs.

These same studies consistently associated AKI in the first 100 days after HSCT with lower overall survival—Kanduri et al. (2020) published a meta-analysis [13] showing pooled odds ratios of 3-month mortality and 3-year mortality among patients undergoing HSCT with AKI were 3.05 (95% CI 2.07–4.49) and 2.23 (95% CI 1.06–4.73), respectively, with higher mortality in more severe stages. Also, Gutierrez et al. [14] and Andronesi et al. [18] found an association between AKI in allogeneic HSCTs and progression to CKD in HSCTs. These findings reinforce the importance of identifying patients at risk to implement prevention measures and early approach.

Although many AKI risk factors occurring in the first days after HSCT have been previously identified—such as exposure to nephrotoxic drugs, moderate-to-severe mucositis, shock, sepsis graft versus host disease, veno-occlusive disease [17]—we did not include any variables that would occur after HSCTs in our model. By doing so, we assured a predictive risk score applicable at the beginning of the hospital stay that does not need to be repeated nor updated during the following days.

Our predictive risk score takes into consideration the presence of chronic kidney disease previous to HSCT, the HCT-CI score, the platelet-to-lymphocyte ratio at admission and the hematologic malignancy diagnosis. All these variables used as risk factors for the prediction of AKI are consistent with the previous literature on AKI risk factors for AKI in general.

Chronic kidney disease is an extensively known risk factor for AKI [27,28,29]. Chronic kidney disease results in a state of constant relative hypoxia with reduced numbers of peritubular capillaries, increased deposition of collagen, myofibroblast proliferation, increased activation of the renin–angiotensin system and reduced numbers of glomeruli, leading to hyperfiltration and higher tubular oxygen consumption of the corresponding tubules [30]. These aspects combined with the chronic leukocyte infiltration and pro-inflammatory environment of chronic kidney disease result in reduced renal reserve and maladaptation, loss of autoregulation and abnormal vasodilation, which represent the perfect conditions for enhanced susceptibility to developing AKI.

The independent association of an HCT-CI score higher than two points and AKI underscores the importance of previous comorbidities in the context of AKI. This comorbidity index takes into consideration the previous history of cardiovascular, pneumological, gastrointestinal, nephrological, rheumatological, oncological and psychiatric complications. Many of these conditions have been associated with AKI in different clinical scenarios. HCT-CI provides information about the overall, as well as non-relapse, mortality risk a patient is likely to experience after hematopoietic cell transplantation. Its application as a predictor for AKI is thus facilitated for its already worldwide use.

Platelet-to-lymphocyte ratio is a novel inflammatory marker revealing shifts in platelet and lymphocyte counts due to acute inflammatory and prothrombotic states, which has been used in several clinical contexts for predicting inflammation and mortality. It is calculated by dividing platelet count by lymphocyte count, which makes it a simple, inexpensive and rapid marker. It has been associated with worse overall survival in various solid tumors [31], with higher mortality in patients with acute heart failure [32], in septic patients [33], and in patients with rheumatic diseases [34]. It has also been reported an association between platelet-to-lymphocyte ratio and worse prognosis of septic AKI patients [35].

In our study, patients with multiple myeloma had a lower risk of developing AKI compared to patients with either lymphoma or leukemia. We believe this aspect may be related to lower treatment burden previous to HSCTs in patients with multiple myeloma, where HSCT is part of first-line treatment. Patients with lymphoma or leukemia were often exposed to high-dose chemotherapy regimens that are related to higher nephrotoxicity and the performance of HSCTs is often used when the disease relapses or progresses despite chemotherapy regimens. In contrast to myeloma, patients with lymphoma or leukemia were often exposed to high-dose chemotherapy regimens and transplanted in a relapsed/refractory setting, which contributed to nephrotoxicity. The authors consider this observation clinically valuable since myeloma patients are usually considered to be a high-risk population for AKI compared to other hematologic malignancies. However, this may not necessarily be the case during hospitalization for an autograft—a setting where at least a partial response to pre-transplant therapy is mandatory.

The main limitation of our study relates to its single-center retrospective nature and consequent limitation in the generalization of our results. Similarly, the score was developed and tested in the same group of patients, which might have overestimated its overall performance. It is expected that its C-statistic should be different in other clinical cohorts. Despite its limitation, we believe important strengths should be enlightened. Unlike most studies on AKI in HSCTs (and on AKI in general), we used both creatinine and urinary output criteria for the KDIGO classification. This aspect contributes to a more accurate diagnosis of AKI and consequently higher precision and internal validity of our results. The fact that we only used variables available at the time of hospital admission allows the score to be calculated at a single point in time and before undergoing the HSCT, which makes it more user friendly. The inclusion of patients undergoing both autologous and allogeneic HSCTs allows the application of this predictive risk score to a wider HSCT population. Also, our score was created based on clinical characteristics and laboratory results that are easily assessed, inexpensive and are currently part of the evaluation of every patient eligible for an HSCT in all countries that offer this procedure. This aspect makes this score accessible to all clinicians at no additional cost.

Our study provides novelty in two aspects: (1) by being, to the best of our knowledge, the first study proposing a predictive risk score for AKI in patients undergoing HSCTs considering variables available at the time of hospital admission; (2) by being the first study to establish platelet-to-lymphocyte ratio as a risk factor for AKI. This score deserves further validation in multi-center prospective studies.

We believe the introduction of this kind of tool in clinical practice is crucial, as it allows the implementation of preventive measures and earlier diagnosis in patients with a higher risk of developing a complication known to have the worst prognostic impact on overall survival in this population.

## 5. Conclusions

In our study including 433 patients, AKI as defined by the KDIGO classification using both creatinine and urinary output criteria affected more than half of the patients undergoing HSCTs considering the first 100 days after the procedure. Considering data available before undergoing an HSCT, this complication is known to be related to lower overall survival and was associated with patients who already had chronic kidney disease, patients who presented two or more points in the HCT-CI score, patients with the underlying diagnosis of lymphoma or leukemia and patients with a platelet-to-lymphocyte ratio at hospital admission ≥171.9. Considering these findings, we developed a new calculated risk score to predict AKI in patients with hematologic malignancies undergoing an HSCT, which combines clinical and laboratorial markers available at the time of the procedure that are easily assessed and inexpensive. The development of predictive risk scores is very important—identifying patients at high risk is an essential step towards selecting those who might benefit from specific prevention measures and confers higher awareness for an earlier diagnosis.

This score should be validated with multi-center prospective studies.

## Figures and Tables

**Figure 1 cancers-15-03720-f001:**
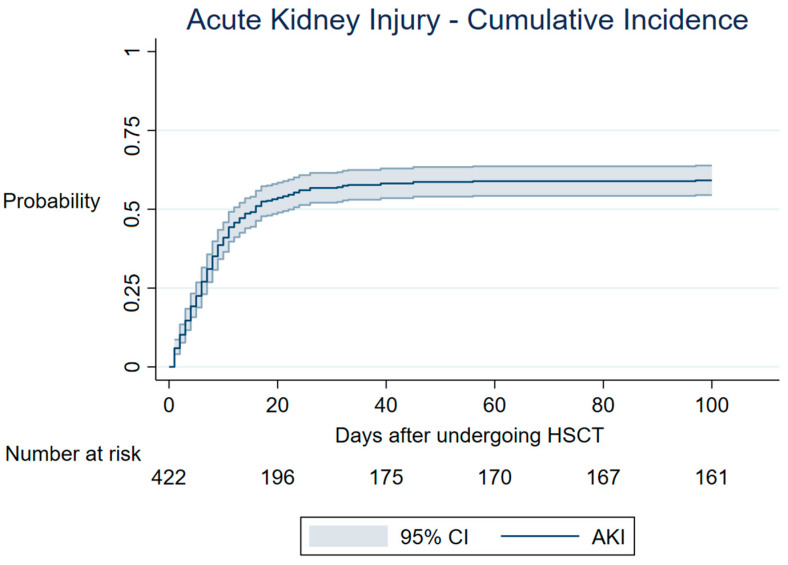
Cumulative incidence of AKI post-HSCT. Cumulative incidence function of AKI according to the KDIGO classification using serum creatinine rise criteria and urinary output criteria. Death was considered a competing event. HSCT—hematopoietic stem cell transplant; CI—cumulative incidence; AKI—acute kidney injury.

**Figure 2 cancers-15-03720-f002:**
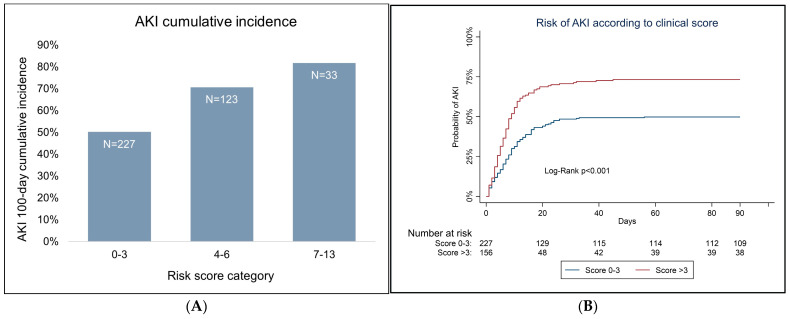
(**A**) AKI cumulative incidence at 100 days according to different score categories. (**B**) AKI distribution curves by score category (score 0–3 and score >3).

**Table 1 cancers-15-03720-t001:** Patients’ baseline characteristics and transplant-related variables.

Patients Characteristics	Category	n (%)	P50	P25	P75
Age at transplant (years)			50.2	36.0	59.5
Gender	Male	236 (55.9)			
	Female	186 (44.1)			
Race	Caucasian	385 (91.4)			
	Non-Caucasian	37 (8.6)			
BMI (Kg/m^2^)			24.6	21.9	27.8
HCT-CI	0–1	353 (84.3)			
	≥2	66 (15.8)			
Hypertension		88 (20.9)			
Diabetes mellitus		26 (6.2)			
Congestive heart failure		23 (5.5)			
Chronic kidney disease		27 (6.4)			
Hematologic diagnosis	Leukaemia	164 (38.8)			
	Lymphoma	115 (27.3)			
	Multiple myeloma	143 (33.9)			
Type of HSCT	Autologous	258 (61.1)			
	Allogeneic	164 (38.9)			
Type of donor	Self	258 (61.1)			
	Related	92 (21.8)			
	Not related	72 (17.1)			
Previous radiotherapy	yes	82 (19.5)			
basal eGFR (ml/min/1.73 m^2^)			107.3	94.3	122.1
Conditioning regimen	Myeloablative	305 (72.2)			
	Non-myeloablative	117 (27.8)			
Graft source	Peripheral blood	389 (92.2)			
	Bone marrow	33 (7.8)			
GVHD prophylaxis	CsA + MMF	117 (27.7)			
	CsA + MTX	47 (11.1)			
	None	258 (61.1)			
At hospital admission day:					
Hemoglobin (gr/dL)			11.6	10.2	12.6
Leukocytes (cells/mm^3^)			4920	3500	6860
Neutrophils (cells/mm^3^)			2960	1850	4420
Platelets (/μL)			179,000	127,000	245,000
Urea (mg/dL)			33	27	41
Uric acid (mg/dL)			5	4	6
Calcium (mg/dL)			9	8.8	10
Phosphate (mg/dL)			4	3.2	4.1
Reactive C protein (mg/dL)			0.49	0.15	2
Lactate dehydrogenase (U/L)			339	291	426
Albumin (mg/dL)			4	3.7	4.5
Alanine transaminase (U/L)			22	15	37
Total bilirubin (mg/dL)			0.48	0.36	0.61
Platelet-to-lymphocyte ratio			159.9	94.1	265.4

SD—standard deviation; P50—median; P25—25th percentile; P75—75th percentile; P25 BMI—body mass index; HCT-CI—hematopoietic stem cell transplant comorbidity index; eGFR—estimated glomerular filtration rate; CsA—cyclosporine; MMF—mycophenolate mofetil; MTX—methotrexate.

**Table 2 cancers-15-03720-t002:** Univariable analysis for AKI including all variables.

Patient’s Characteristics	HR Estimate	95% CI		*p*-Value
		Lower Limit	Upper Limit	
Age at transplant (years)	1.00	0.99	1.01	0.830
Gender (female versus male)	1.05	0.83	1.35	0.654
Race (Caucasian versus non-Caucasian)	0.90	0.58	1.40	0.634
BMI (Kg/m^2^)	1.02	1.00	1.05	0.105
HCT-CI (score < 2 versus score ≥ 2)	1.69	1.27	2.25	<0.001
Hypertension	1.15	0.86	1.54	0.335
Diabetes mellitus	1.29	0.82	2.02	0.269
Congestive heart failure	1.41	0.91	2.20	0.128
Chronic kidney disease	2.11	1.36	3.27	0.001
Hematologic diagnosis:				
Multiple myeloma versus lymphoma	1.51	1.10	2.08	0.011
Multiple myeloma versus leukemia	1.44	1.07	1.92	0.015
Leukemia versus lymphoma	1.05	0.78	1.41	0.728
Multiple myeloma versus (lymphoma + leukemia)	1.46	1.12	1.90	0.005
Type of HSCT (allogeneic versus autologous)	0.84	0.66	1.08	0.169
Type of donor (related versus unrelated)	1.14	0.86	1.52	0.368
Previous radiotherapy	1.09	0.81	1.47	0.573
Conditioning regimen (non-myeloablative versus myeloablative)	1.27	0.84	1.95	0.260
basal eGFR (ml/min/1.73 m^2^)	0.99	0.99	1.00	0.012
Graft source (peripheral blood versus bone marrow)	1.36	0.88	2.12	0.170
GVHD prophylaxis (methotrexate versus others)	1.03	0.73	1.46	0.849
At hospital admission day:				
Hemoglobin (gr/dL)	1.01	0.94	1.09	0.752
Leukocytes * (cells/mm^3^)	1.07	1.04	1.09	<0.001
Neutrophils * (cells/mm^3^)	1.06	0.88	1.26	0.547
Lymphocytes * (cells/mm^3^)	1.23	1.13	1.34	<0.001
Platelets * (/μL)	1.01	0.99	1.02	0.841
Urea (mg/dL)	1.01	1.00	1.02	0.019
Uric acid (mg/dL)	1.01	0.97	1.04	0.662
Calcium (mg/dL)	1.06	0.85	1.31	0.598
Phosphate (mg/dL)	0.92	0.77	1.10	0.367
Reactive C protein ** (mg/dL)	1.02	1.01	1.03	<0.001
Lactate dehydrogenase ** (U/L)	1.05	1.03	1.07	<0.001
Albumin (mg/dL)	1.00	0.96	1.04	0.847
Alanine transaminase (U/L)	1.00	0.99	1.00	0.081
Total bilirubin (mg/dL)	0.91	0.62	1.32	0.611
Platelet-to-lymphocyte ratio	1.00	1.00	1.00	<0.001

*—for each rise of 1000; **—for each rise of 10; BMI—body mass index; HCT-CI—hematopoietic stem cell transplant comorbidity index; Nr—number; HSCT—hematopoietic stem cell transplant; eGFR—estimated glomerular filtration rate; GVHD—grafts versus host disease.

**Table 3 cancers-15-03720-t003:** Univariable analysis for AKI including the selected variables (*p* < 0.200 in Table 2) after applying the Liu index and establishing the optimal cut-off point.

Patient’s Characteristics	HR Estimate	95% CI		*p*-Value
		Lower Limit	Upper Limit	
BMI (>24.5 Kg/m^2^)	1.20	0.94	1.53	0.147
HCT-CI (socre < 2 versus score ≥ 2)	1.69	1.27	2.25	<0.001
Congestive heart failure	1.41	0.91	2.2	0.128
Chronic kidney disease	2.11	1.36	3.27	0.001
Hematologic diagnosis (multiple myeloma versus (lymphoma + leukemia)	1.46	1.12	1.91	0.005
Type of HSCT (allogeneic versus autologous)	0.84	0.66	1.08	0.169
basal eGFR (>107.2 mL/min/1.73 m^2^)	0.89	0.7	1.13	0.342
Graft source (peripheral blood versus bone marrow)	1.36	0.88	2.12	0.170
Leukocytes (>5330 cells/mm^3^)	1.05	0.82	1.34	0.696
Lymphocytes (>1100 cells/mm^3^)	0.89	0.7	1.13	0.334
Urea (>30 mg/dL)	1.15	0.9	1.48	0.263
Reactive C protein (>0.47 mg/dL)	1.20	0.92	1.55	0.176
Lactate dehydrogenase (>339 U/L)	1.26	0.98	1.61	0.066
Alanine transaminase (>22 U/L)	1.02	0.8	1.31	0.853
Platelet-to-lymphocyte ratio (>171.9)	1.26	0.97	1.62	0.078

BMI—body mass index; HCT-CI—hematopoietic stem cell transplant comorbidity index; HSCT—hematopoietic stem cell transplant; eGFR—estimated glomerular filtration rate.

**Table 4 cancers-15-03720-t004:** Multivariable analysis for AKI with Score points.

Patient’s Characteristics	HR Estimate	95% CI		*p*-Value	Score Points
		Lower Limit	Upper Limit		
HCT-CI (reference category < 2)	1.47	1.08	2.00	0.013	3
Chronic kidney disease	2.10	1.31	3.36	0.002	4
Hematologic diagnosis (reference category multiple myeloma)	1.69	1.26	2.25	<0.001	3
Platelet-to-lymphocyte ratio (reference category < 171.9)	1.43	1.1	1.86	0.008	3
Multivariable model C-Statistic = 0.71					
Score C-Statistic = 0.70					

HCT-CI—hematopoietic stem cell transplant comorbidity index.

## Data Availability

The data underlying this article will be shared on reasonable request to the corresponding author.

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
