# Peer review of "Predictive Risk Score for Acute Kidney Injury in Hematopoietic Stem Cell Transplant"

_cancers, 2023, doi:10.3390/cancers15143720_

Round 1

Reviewer 1 Report

1. I couldn't find the abstract in the text. Check this one in the article and correct it:
Simple Summary: The incidence and prevalence of hematological malignancies are increasing throughout the world and hematopoietic stem cell transplant contributes to significantly better outcomes. Acute Kidney Injury is a frequent complication of Hematopoietic Stem Cell Transplant and has known implications in overall survival. We calculated a simple, easily assessed, inexpensive predictive risk score to identify patients with hematological malignancies undergoing Hematopoi- 17 etic Stem Cell Transplant at risk for AKI.
Abstract: A single paragraph of about 200 words maximum. For research articles, abstracts should give a pertinent overview of the work. We strongly encourage authors to use the following style of structured abstracts, but without headings: (1) Background: Place the question addressed in a broad context and highlight the purpose of the study; (2) Methods: briefly describe the main methods or treatments applied; (3) Results: summarize the article’s main findings; (4) Conclusions: indicate the main conclusions or interpretations. The abstract should be an objective representation of the article and it must not contain results that are not presented and substantiated in the main text and should 25 not exaggerate the main conclusions.

2. The conclusion is really short. Please provide a suitable conclusion.

3. What specific factors contribute to the increasing incidence and prevalence of hematological malignancies worldwide, and how does hematopoietic stem cell transplant (HSCT) play a significant role in improving disease-free survival and overall survival?

Minor editing of the English language is required.

Author Response

Dear reviewer 1,

Thank you very much for your time on reviewing our manuscript. We made the suggested changes directly in the attached revised manuscript. We will answer your questions following the order

  1. I couldn't find the abstract in the text. Check this one in the article and correct it:

We apologize for this mistake; we whereby share the reviewed abstract:

Hematopoietic Stem Cell Transplant (HSCT) is an important treatment option for hematologic malignancies. Acute Kidney Injury is a common complication in HSCT and is related to worse outcomes. We aimed to create a predictive risk score for AKI in HSCT considering variables available at the time of the transplant. We performed a retrospective cohort study. AKI was defined by the KDIGO classification using creatinine and urinary output criteria. We used survival analysis with competing events. Continuous variables were dichotomized according to the Liu index. A multivariable analysis was performed with a backward stepwise regression.  Harrel’s C-Statistic was used to evaluate the performance of the model. Points were attributed according to Hazard Ratio. The Liu index was used to establish the optimal cut-off. We included 422 adult patients undergoing autologous (61.1%) or allogeneic (38.9%) HSCT for Multiple Myeloma (33.9), Lymphoma (27.3), and Leukemia (38.8%). AKI cumulative incidence was 59.1%. Variables eligible for the final score were: Hematopoietic Cell Transplant Comorbidity index ≥2 (HR:1.47,95%CI:1.08-2.006;p=0.013), Chronic Kidney Disease (HR:2.10,95%CI:1.31-3.36;p=0.002), Lymphoma or Leukemia (HR:1.69,95%CI:1.26-2.25;p<0.001) and Platelet-to-lymphocyte ratio > 171.9 (HR:1.43,95%CI:1.10-1.86;p=0.008). This is the first predictive risk score for AKI in patients undergoing HSCT and the first study where the Platelet-to-lymphocyte ratio is independently associated with AKI.

  1. The conclusion is really short. Please provide a suitable conclusion.

We reformulated our conclusion and hope to be more appropriate.

In our study including 433 patients AKI defined by the KDIGO classification using both creatinine and urinary output criteria affected more than half of the patients undergoing HSCT considering de first 100 days after the procedure.

Considering data available before undergoing HSCT, this complication known to be related to lower overall survival was associated to patients who already had chronic kidney disease, patients who presented two or more points in the HCT-CI score, patients with the underlying diagnosis of lymphoma or leukemia and patients with a platelet-to-lymphocyte ratio at hospital admission ≥171.9. Considering these findings, we developed a new calculated risk score to predict AKI in patients with hematologic malignancies undergoing HSCT, which combines clinical and laboratorial markers available at the time of the procedure that are easily assessed and inexpensive. The development of predictive risk scores is very important – identifying patients at high risk is an essential step towards selecting those who might benefit from specific prevention measures and confers higher awareness for an earlier diagnosis.

This score should be validated with multi-center prospective studies.

  1. What specific factors contribute to the increasing incidence and prevalence of hematological malignancies worldwide, and how does hematopoietic stem cell transplant (HSCT) play a significant role in improving disease-free survival and overall survival?

We thank you for your question, we have incorporated part of our response in our manuscript in order to complete it.

Hematologic malignancies are registering an increasing incidence and prevalence worldwide (Global Burden of Disease 2019 Cancer Collaboration. Cancer Incidence, Mortality, Years of Life Lost, Years Lived With Disability, and Disability-Adjusted Life Years for 29 Cancer Groups From 2010 to 2019: A Systematic Analysis for the Global Burden of Disease Study 2019. JAMA Oncol. 2022;8(3):420–444) Several factors contribute to this evidence. Diagnostic standardization according to universal classification systems such as the World Health Organization Classification of Tumour and Haematopoietic and Lymphoid Tissues (Alaggio R, Amador C, Anagnostopoulos I, Attygalle AD, Araujo IBO, Berti  et al. The 5th edition of the World Health Organization Classification of Haematolymphoid Tumours: Lymphoid Neoplasms. Leukemia. 2022 Jul;36(7):1720-1748), has resulted in better data collection and higher reporting on regional and international cancer registries. Also, earlier and more precise diagnoses have been a predictable consequence of the evolution of cancer genetic and molecular testing technologies. These two factors have partially overcome underdiagnosis and under-reporting, contributing to a higher incidence of hematologic malignancies. Advanced age is a well-recognized risk factor for cancer development and is partially related to DNA damage accumulation and immunosenescence (Lian, J., Yue, Y., Yu, W. et al. Immunosenescence: a key player in cancer development. J Hematol Oncol 13, 151 (2020)). The aging of the global population has resulted in a steady increase in hematologic malignancies, both lymphoid and myeloid neoplasms (HAEMACARE Working Group. Incidence of hematologic malignancies in Europe by morphologic subtype: results of the HAEMACARE project. Blood. 2010 Nov 11;116(19):3724-34). Higher cancer survivorship rates due to improvements in oncology care have paradoxically created conditions for the development of therapy-related secondary myeloid malignancies (McNerney ME, Godley LA, Le Beau MM. Therapy-related myeloid neoplasms: when genetics and environment collide. Nat Rev Cancer. 2017 Aug 24;17(9):513-527), which tend to occur in long-term survivors.

With the increasing prevalence of hematologic malignancies, the overall number of patients requiring HSCT has also evolved. HSCT is a potentially curative treatment for virtually all hematologic cancers and the therapeutic benefits result from high-dose chemotherapy and the graft-vs-tumour effect that develops after allografting.  Acute leukemias are the most common indications for allogeneic HSCT (Snowden, J.A., Sánchez-Ortega, I., Corbacioglu, S. et al. Indications for haematopoietic cell transplantation for haematological diseases, solid tumours and immune disorders: current practice in Europe, 2022. Bone Marrow Transplant 57, 1217–1239 (2022)). Leukaemias harboring adverse genetic events are at high risk of relapse after intensive chemotherapy, and post-remission allografting is the only available treatment modality that can result in cure or long-term survival(Snowden, J.A., Sánchez-Ortega, I., Corbacioglu, S. et al. Indications for haematopoietic cell transplantation for haematological diseases, solid tumours and immune disorders: current practice in Europe, 2022. Bone Marrow Transplant 57, 1217–1239 (2022)). Multiple Myeloma is the most common adult indication for autologous HSCT(Snowden, J.A., Sánchez-Ortega, I., Corbacioglu, S. et al. Indications for haematopoietic cell transplantation for haematological diseases, solid tumours and immune disorders: current practice in Europe, 2022. Bone Marrow Transplant 57, 1217–1239 (2022)). The steep dose-response curve for high-dose melphalan in patients with Myeloma results in meaningful clinical benefits after HSCT and this treatment modality delays progression and improves median overall survival by approximately 12 months (Rajkumar, SV. Multiple myeloma: 2022 update on diagnosis, risk stratification, and management. Am J Hematol. 2022; 97( 8): 1086- 1107). Relapsed / Refractory Hodgkin (HL) and non-Hodgkin Lymphomas (NHL) are the second most common adult indications for autologous HSCT (Snowden, J.A., Sánchez-Ortega, I., Corbacioglu, S. et al. Indications for haematopoietic cell transplantation for haematological diseases, solid tumours and immune disorders: current practice in Europe, 2022. Bone Marrow Transplant 57, 1217–1239 (2022)  and curability rates are high. The fact that increasing alkylating agent dosing can overcome the resistance of most lymphoma cells defines the biological rationale for the clinical use of autologous HSCT in second-line HL and NHL. In relapsed / refractory diffuse large B cell lymphoma, the most common NHL, the role of HSCT has been prospectively established and compared to chemotherapy alone, 5-year event-free survival and overall survival were significantly superior in the transplant arm (46% and 53% vs 12 and 32%, respectively) (Philip T, Guglielmi C, Hagenbeek A, M.D., Somers R, Lelie H, Bron D.Autologous Bone Marrow Transplantation as Compared with Salvage Chemotherapy in Relapses of Chemotherapy-Sensitive Non-Hodgkin's Lymphoma. N Engl J Med 1995; 333:1540-1545). In HL, a disease of young adults, autologous HSCT is the standard of care in the relapsed / refractory setting and results in a cure rate of 50% (Arai S, Fanale M, DeVos S, Engert A, Illidge T, Borchmann P, Younes A, et al. Defining a Hodgkin lymphoma population for novel therapeutics after relapse from autologous hematopoietic cell transplant. Leuk Lymphoma. 2013 Nov;54(11):2531-3.

Due to several advances in the field, HSCT is becoming more widely applicable and patient outcomes are continuously improving. Indications for this treatment modality are expected to expand and evolve over the following years, which highlights the need for better characterization of short and long-term complications such as renal disease.

In data presented in 2019 by the Worldwide Network for Blood and Marrow Transplantation (WBMT) (Dietger Niederwieser, Helen Baldomero, Yoshiko Atsuta, Mahmoud Aljurf, Adriana Seber, Hildegard T. Greinix et al. One and Half Million Hematopoietic Stem Cell Transplants (HSCT). Dissemination, Trends and Potential to Improve Activity By Telemedicine from the Worldwide Network for Blood and Marrow Transplantation (WBMT). Blood 2019; 134 (Supplement_1): 2035), from 1957-2016 a total of 1.298.897 HSCT (57.1% autologous) procedures had been performed. HSCT activity has been reported from 87 of the 195 World Health Organization (WHO) member states. The global activity/year has been increasing continuously from 10.000/year in 1991 to 82.718 first HSCT/year in 2016, with slightly more autologous (53.5%) than allogeneic HSCT. This represents an increase > 7% per year.

Reviewer 2 Report

The manuscript addresses a very interesting topic of developing of the predictive risk score for Acute Kidney Injury in patients undergoing Hematopoietic Stem Cell Transplantation.

Although HSCT offers a cure for many malignancies, it can also be associated with serious treatment complications, including AKI. The objectives of the work are clearly presented. The study design is presented in the Materials and Methods section. The study, although retrospective and single-centre, was conducted on a large group of patients, which allows for reliable results of statistical analyzes. Statistical methods were correctly presented and used for analyses.

The results are clearly presented in the form of tables and figures.

In the discussion, the authors presented the possibilities of using the predictive AKI risk index in patients treated with HSCT, while emphasizing the limitations of the study, which are associated with its single-center, retrospective nature.

References are mostly publications that appeared in the last 10 years.

In conclusion, the work is an important voice in the discussion on a significant complication among patients treated with HSCT, which is AKI, although the results obtained need to be confirmed by multicentre studies on a larger group of patients.

Author Response

Dear reviewer 2,

Thank you very much for your time on reviewing our manuscript. We truly appreciate your positive feedback and will continue our research on this field.

Reviewer 3 Report

This paper presents a predictive risk score for acute kidney injury within 100 days of HSCT, and appropriately considers only factors that are easy to collect and are known at the time of HSCT.  Overall cumulative incidence of AKI in their single center cohort was 59%.  The risk score that they derived identified small groups of patients with a predicted 100 day cumulative incidence of AKI as low as 35%, and as high as 82%.  Most patients had a moderate risk score with predicted 100 day cumulative incidence of 50-70%, near the overall CI of 59%.

Predictive risk scores attempt to identify a small subset of the top predictors for a condition, weight them, and group patients into a small number of categories that stratify risk.  This involves a tradeoff between model accuracy and simplicity; there is necessarily a considerable amount of rounding error introduced due to categorization which is inherent to risk score development.  Thus, the statistical methods need not be optimal, as long as the resulting risk model is clinically useful and also accurate when applied to new patient cohorts.  Since no validation was reported in this paper, this remains an open question.

That said, statisticians have identified serious problems with two of the methods used in this paper - univariable selection combined with an automatic variable selection algorithm, and using cutpoints for continuous variables.  So-called "optimal" cutpoints rarely replicate across studies and produce statistical models that are biologically implausible - rarely does the risk of anything make a "quantum leap" at a cutpoint of a continuous variable.  For more information, please see anything written on these topics by Frank Harrell, whose c-index is cited in this paper. (Some links to more information are provided on this webpage: https://biostat.app.vumc.org/wiki/Main/ManuscriptChecklist).  Because of what I wrote in the previous paragraph, the analysis does not need to be redone, but I would like to share this information with the authors.

Some specific suggestions that could improve the paper:

1. There is no abstract.

2. In Table 1, it is unclear which numbers represent a mean or median.  It would be better to pick one set of summary statistics and use it consistently throughout the table.  I like P25, P50, and P75, but please just make it consistent.

3. In Figure 1, please provide a label for the numbers in parentheses in the "number at risk" line.

4. In Table 2, please write the applicable category in the row label, for example: Male (vs female), as you did for hematologic diagnosis.  For variables with more than two categories, it is unclear what is compared; for example, the reference category for donor type is related, but is that compared to self or unrelated donors (or both)?

Many of the biomarkers have a confidence interval of (1.00, 1.00), because the model is estimating a hazard ratio for a one-unit difference in the biomarker.  Please change the scale to get meaningful hazard ratios and confidence intervals.  Alternatively, you could pick two reference values (like the 25th and 75th percentile) within each variable's distribution, and report the hazard ratio corresponding to those reference values.

5. In Figure 2, axis labels would make the left panel more intuitive.  Something like "Risk score category" for the x-axis, and "AKI 100-day cumulative incidence" for the y-axis.  Likewise, the axis labels on the right panel could be improved.

Please show the entire risk score distribution; 1, 2, and 5 are missing.

Figure 2 is a nice concept but could be improved if the left and right panels were consistent with each other, by showing the same categories in each plot.

6. Line 134 states that cumulative incidence was 59% after autologous HSCT.  Should this be all HSCT or is it really just autologous?

7. From the number at risk table in Figure 2, it can be deduced that 227+156=383 patients have a risk score, which is less than the total of 422.  This is important information to note - i.e. the percentage of patients who had an unknown risk score and why.

8. It is well known that predictive models tend to be more accurate when applied to the data that was used to develop the model, compared to other data such as a "test" or "validation" cohort.  Since a validation cohort was not used, the discussion should at least acknowledge that this risk model would likely be less accurate (lower c-index, less risk separation between categories)  when applied to other cohorts.

9. Figure 2 shows that most patients get a risk score that corresponds to a 52% or 71% probability of AKI.  Since the overall probability of AKI is 59%, how useful is this risk score to clinicians and patients?  Would recommended treatment change based on whether a patient has approximately a 50% probability of AKI vs 70%?  Discussion of this would be useful to add to the discussion section.

The language is very understandable.  There are some misspellings and formatting inconsistencies; for example, "category" is misspelled in the tables.

Author Response

Dear Reviewer 3,

Thank you very much for your time on reviewing our manuscript, particularly for your comments on the statistics presentation and approach. We made the possible suggested changes directly in the attached revised manuscript. We will try to answer your questions following the order:

There is no abstract.

We apologize for this mistake, we realized the last version was not upload; we whereby share the abstract:

Hematopoietic Stem Cell Transplant (HSCT) is an important treatment option for hematologic malignancies. Acute Kidney Injury is a common complication in HSCT and is related to worse outcomes. We aimed to create a predictive risk score for AKI in HSCT considering variables available at the time of the transplant. We performed a retrospective cohort study. AKI was defined by the KDIGO classification using creatinine and urinary output criteria. We used survival analysis with competing events. Continuous variables were dichotomized according to the Liu index. A multivariable analysis was performed with a backward stepwise regression.  Harrel’s C-Statistic was used to evaluate the performance of the model. Points were attributed according to Hazard Ratio. The Liu index was used to establish the optimal cut-off. We included 422 adult patients undergoing autologous (61.1%) or allogeneic (38.9%) HSCT for Multiple Myeloma (33.9), Lymphoma (27.3), and Leukemia (38.8%). AKI cumulative incidence was 59.1%. Variables eligible for the final score were: Hematopoietic Cell Transplant Comorbidity index ≥2 (HR:1.47,95%CI:1.08-2.006;p=0.013), Chronic Kidney Disease (HR:2.10,95%CI:1.31-3.36;p=0.002), Lymphoma or Leukemia (HR:1.69,95%CI:1.26-2.25;p<0.001) and Platelet-to-lymphocyte ratio > 171.9 (HR:1.43,95%CI:1.10-1.86;p=0.008). This is the first predictive risk score for AKI in patients undergoing HSCT and the first study where the Platelet-to-lymphocyte ratio is independently associated with AKI.

In Table 1, it is unclear which numbers represent a mean or median.  It would be better to pick one set of summary statistics and use it consistently throughout the table.  I like P25, P50, and P75, but please just make it consistent.

We proceeded to the suggested changes and chose P25, P50, and P75 for representation. The new table is in the revised manuscript

In Figure 1, please provide a label for the numbers in parentheses in the "number at risk" line.

The numbers in parentheses concerned the number of AKI episodes between respective 20-day intervals but now that you mentioned those numbers, we believe it is difficult to label them and maintained them in the picture without rising confusion to the reader so we decided to remove them and leave only number at risk.

In Table 2, please write the applicable category in the row label, for example: Male (vs female), as you did for hematologic diagnosis.  For variables with more than two categories, it is unclear what is compared; for example, the reference category for donor type is related, but is that compared to self or unrelated donors (or both)?

We proceeded to the suggested changes and the new table is in the revised manuscript.

Many of the biomarkers have a confidence interval of (1.00, 1.00), because the model is estimating a hazard ratio for a one-unit difference in the biomarker.  Please change the scale to get meaningful hazard ratios and confidence intervals.  Alternatively, you could pick two reference values (like the 25th and 75th percentile) within each variable's distribution, and report the hazard ratio corresponding to those reference values.

We proceeded to the modifications and present it in the revised table 2.

Please show the entire risk score distribution; 1, 2, and 5 are missing.

The graph is now changed according to the reviewer’s suggestion.

Figure 2 is a nice concept but could be improved if the left and right panels were consistent with each other, by showing the same categories in each plot.

The authors agree it would be easier to interpret the data with better consistency between panels. You can now find in Figure 2 the new risk categories for the bar graph.

Line 134 states that cumulative incidence was 59% after autologous HSCT.  Should this be all HSCT or is it really just autologous?

This cumulative incidence concerns all HSCT, we apologize for the mistake, it is corrected in the text.

Figure 2 shows that most patients get a risk score that corresponds to a 52% or 71% probability of AKI.  Since the overall probability of AKI is 59%, how useful is this risk score to clinicians and patients? 

We understand the point focused. Still, we believe that a 20% risk difference is clinically very significative. Even 33 patients with 81% of risk representing almost 9% of the total of patients is not neglectable.  

In Figure 2, axis labels would make the left panel more intuitive.  Something like "Risk score category" for the x-axis, and "AKI 100-day cumulative incidence" for the y-axis.  Likewise, the axis labels on the right panel could be improved.

The authors agree with the suggestions concerning Figure 2. The axis labels were improved and it is now easier to interpret the results.

From the number at risk table in Figure 2, it can be deduced that 227+156=383 patients have a risk score, which is less than the total of 422.  This is important information to note - i.e. the percentage of patients who had an unknown risk score and why.

The authors appreciate the reviewer’s attention to detail. Only 383 patients presented a calculated risk score due to missing values of the platelet-to-lymphocyte ratio which was included in the model. As this represents less than 10% of missing values, no techniques of imputation were used. This information is now included in the methods section.

 It is well known that predictive models tend to be more accurate when applied to the data that was used to develop the model, compared to other data such as a "test" or "validation" cohort.  Since a validation cohort was not used, the discussion should at least acknowledge that this risk model would likely be less accurate (lower c-index, less risk separation between categories) when applied to other cohorts.

The authors agree with this observation. Every score indeed needs extensive validation in multiple cohorts to reach reproducible model performance. Therefore, you can now find in the discussion an acknowledgment of that fact.

Reviewer 4 Report

Abstract need revision. Part of the abstract contain cut and paste of instruction to the authors. Please check.

Authors should mention whether all the experiments conducted after HSCT for AKI were the part of the standard protocol/evaluation of post-HSCT or they are separate experiments. If so, does the authors taken consent of the patients and Institutional Ethics Committee approval?

In conclusion, authors demonstrated that risk of development of AKI is low with multiple myeloma but high in Lymphoma and Leukemia. What about other variables like Autologous/allogenic transplant, male or female etc.

However, results are quite informative.  

Manuscript need minor correction in language. 

Author Response

Dear reviewer 4,

Thank you very much for your time on reviewing our manuscript. We made the suggested changes directly in the attached revised manuscript. We will answer your questions following the order

  1. Abstract need revision. Part of the abstract contain cut and paste of instruction to the authors. Please check.

We apologize for this mistake; we whereby share the reviewed abstract:

Hematopoietic Stem Cell Transplant (HSCT) is an important treatment option for hematologic malignancies. Acute Kidney Injury is a common complication in HSCT and is related to worse outcomes. We aimed to create a predictive risk score for AKI in HSCT considering variables available at the time of the transplant. We performed a retrospective cohort study. AKI was defined by the KDIGO classification using creatinine and urinary output criteria. We used survival analysis with competing events. Continuous variables were dichotomized according to the Liu index. A multivariable analysis was performed with a backward stepwise regression.  Harrel’s C-Statistic was used to evaluate the performance of the model. Points were attributed according to Hazard Ratio. The Liu index was used to establish the optimal cut-off. We included 422 adult patients undergoing autologous (61.1%) or allogeneic (38.9%) HSCT for Multiple Myeloma (33.9), Lymphoma (27.3), and Leukemia (38.8%). AKI cumulative incidence was 59.1%. Variables eligible for the final score were: Hematopoietic Cell Transplant Comorbidity index ≥2 (HR:1.47,95%CI:1.08-2.006;p=0.013), Chronic Kidney Disease (HR:2.10,95%CI:1.31-3.36;p=0.002), Lymphoma or Leukemia (HR:1.69,95%CI:1.26-2.25;p<0.001) and Platelet-to-lymphocyte ratio > 171.9 (HR:1.43,95%CI:1.10-1.86;p=0.008). This is the first predictive risk score for AKI in patients undergoing HSCT and the first study where the Platelet-to-lymphocyte ratio is independently associated with AKI.

  1. Authors should mention whether all the experiments conducted after HSCT for AKI were the part of the standard protocol/evaluation of post-HSCT or they are separate experiments. If so, does the authors taken consent of the patients and Institutional Ethics Committee approval?

All the collected data was available previous to the study. This study was approved by the local Ethical Committee – Comissão de Ética do Centro Académico de Medicina de Lisboa on the 13th of October of 2020 (reference number 334/20) in agreement with institutional guidelines. Informed consent was waived by the Ethical Committee due to the retrospective and non-interventional nature of the study.

  1. In conclusion, authors demonstrated that risk of development of AKI is low with multiple myeloma but high in Lymphoma and Leukemia. What about other variables like Autologous/allogenic transplant, male or female etc.

In table 2 we present univariable analysis for AKI including all collectable variables, we analyzed the demographic characteristics (age, gender, race, BMI, etc) and HSCT characteristics (autologous vs allogeneic, type of donor, conditioning regimen). We did not find statistically significant differences between the groups. As pointed out by the hazard ratios, confidence intervals and p-values presented in this table.

Reviewer 5 Report

Rodriguez and colleagues wanted to present a predictive risk score for Acute Kidney Injury in Hematopoietic Stem Cell Transplant. The mission is very interesting but in my opinion is not well explained.

1. Please check the abstract and formulate a paragraph in which you well describe the predictive risk score that you have calculated and how

2. Table 1 is not complete. Please check the missing data. For example Female number

3. Par. 2.4. Why have you analyzed only autologous data for AKI incidence?  What is the data for allogenic HSCT? What is the relationship with the incidence, the type of transplant and the pathologies? There is a difference between male or female?

4. What about the age of the transplanted? The is a bad relationship with the age or not? 

5. Please explain better the results of the tables that are not so easy to understand. 

6. How you have calculated risk score? I have not understand

Please add the missing parts and reformulate the text well so that it responds well to the expectations of the title and it is well understood how you want to bring something new to a topic on which there is a lot of literature, namely predict risk score. 

Author Response

Dear Reviewer 5,

Thank you very much for your time on reviewing our manuscript. We made the possible suggested changes directly in the attached revised manuscript. We will try to answer your questions following the order:

  1. Please check the abstract and formulate a paragraph in which you well describe the predictive risk score that you have calculated and how

We reformulated the abstract´s paragraph concerning the predictive score considering the limitation of words for this section.

  1. Table 1 is not complete. Please check the missing data. For example Female number

Considering the number of variables, we thought it would be more pleasant to read a not so full table. When we consider a number and a percentage for males, we are assuming the reader understands that the remaining percentage to reach 100% is for females. Though we reformulated table 1 in order to include all the information.

  1. 2.4. Why have you analyzed only autologous data for AKI incidence?  What is the data for allogenic HSCT? What is the relationship with the incidence, the type of transplant and the pathologies? There is a difference between male or female?

We apologize for the mistake, the 59.1% of AKI cumulative incidence concerns total AKI incidence of the study, including autologous and allogeneic transplant.

The incidence is expressed on table 1 -  the type of transplant (autologous 61.1% and allogeneic 38.9%), Multiple Myeloma(33.9%), Lymphoma(27.3%), and Leukemia(38.8%) as well as the sex (male 55.9% and female 44.1%).

The relationship (which we assume you refer to AKI) is expressed on table 2,where you can find the univariable analysis presenting Hazard Ratio estimate, p-value and 95% confidence interval. 

  1. What about the age of the transplanted? The is a bad relationship with the age or not?

In our univariable analysis for AKI, age did not show any significative impact on AKI. The result  is expressed on table 2.

  1. How you have calculated risk score? I have not understand

First, we analysed AKI as suggested by the European Group for Blood and Marrow Transplantation statistical approach guidelines (Iacobelli S, Committee ES. Suggestions on the use of statistical methodologies in studies of the European Group for Blood and Marrow Transplantation. Bone Marrow Transpl. 2013;48:S1–37).

In order to do so, we used survival analysis methods considering competing events – we considered death a competing risk event. For that analysis we used the Fine and Gray method(Fine, J. P., and R. J. Gray. 1999. A proportional hazards model for the subdistribution of a competing risk. Journal of the American Statistical Association 94: 496–509) to calculate the cumulative incidence of AKI and perform univariable analysis of factors predicting AKI and we presented results in table 2.

Considering that continuous variables were not feasible for a score, we calculated the optimal cut-off point  for all continuous variables with a p <0.200 in univariable analysis (p< 0.200 was our criteria for admission to the multivariable model creation). To find that cut-off we used the Liu index (Liu Index defines the optimal cut-off as the point which maximizes the product of the sensitivity and specificity - Liu X. Classification accuracy and cut point selection. Statistics in medicine. 2012;31(23):2676-86.). These results are presented in table 3.

Then, also trough the Fine and Gray method, we created our multivariable model was through a backward stepwise regression (entry criteria – p<0.200 as mentioned above).

That model is presented in table 4. Harrel’s C-Statistic was used to evaluate the performance of the model (also shown in table4)  and a risk score was created by attributing points corresponding to the nearest integer of 2 times each covariate’s Hazard Ratio (last column onin table 4). The Liu index was used to establish the optimal cut-off for the clinical risk score.

A Log-Rank test and respective reverse Kaplan-Meier curves were used to compare event distributions of the resulting score categorical variable.

Round 2

Reviewer 3 Report

I thank the authors for their detailed responses and for making suggested improvements to their paper.  No additional comments.

Reviewer 5 Report

The authors have answered to all my revisions and to the contribute of the other reviewer and to the author's implementation, the paper have now another sound. Thanks to the authors.